# T Cell Based Immunotherapy for Cancer: Approaches and Strategies

**DOI:** 10.3390/vaccines11040835

**Published:** 2023-04-13

**Authors:** Muzamil Y. Want, Zeenat Bashir, Rauf A. Najar

**Affiliations:** 1Department of Immunology, Division of Translational Immuno-Oncology, Roswell Park Comprehensive Cancer Center, Buffalo, NY 14263, USA; 2Department of Chemistry and Biochemistry, Canisius College, Buffalo, NY 14208, USA; bashirz@canisius.edu; 3Department of Pediatrics, Lung Biology and Disease Program, University of Rochester School of Medicine and Dentistry, Rochester, NY 14642, USA

**Keywords:** T cells, immunotherapy, neoantigens, cancer antigens, TCR engineering, HLA restriction, chimeric antigen receptor, CAR NK cells, CAR MIAT

## Abstract

T cells are critical in destroying cancer cells by recognizing antigens presented by MHC molecules on cancer cells or antigen-presenting cells. Identifying and targeting cancer-specific or overexpressed self-antigens is essential for redirecting T cells against tumors, leading to tumor regression. This is achieved through the identification of mutated or overexpressed self-proteins in cancer cells, which guide the recognition of cancer cells by T-cell receptors. There are two main approaches to T cell-based immunotherapy: HLA-restricted and HLA-non-restricted Immunotherapy. Significant progress has been made in T cell-based immunotherapy over the past decade, using naturally occurring or genetically engineered T cells to target cancer antigens in hematological malignancies and solid tumors. However, limited specificity, longevity, and toxicity have limited success rates. This review provides an overview of T cells as a therapeutic tool for cancer, highlighting the advantages and future strategies for developing effective T cell cancer immunotherapy. The challenges associated with identifying T cells and their corresponding antigens, such as their low frequency, are also discussed. The review further examines the current state of T cell-based immunotherapy and potential future strategies, such as the use of combination therapy and the optimization of T cell properties, to overcome current limitations and improve clinical outcomes.

## 1. Introduction

Cancer is a major public health challenge, affecting millions of people worldwide. Despite significant advances in cancer therapy, cancer remains a leading cause of death, accounting for approximately 10 million deaths annually. The immune system plays a vital role in protecting the host from malignant cells, and harnessing its potential has been a focus of cancer therapy research for decades. T cells are a critical component of the adaptive immune system, and their ability to recognize and eliminate cancer cells has led to significant advances in cancer immunotherapy. In the mid-twentieth century, the concept of lymphocytes as mediators of anti-tumor surveillance was first proposed by Thomas and Burnett, but it was not until later that T cells were identified as the major mediators of adaptive immunity [1,2,3,4,5,6,7]. Today, the role of T cells in cancer immunotherapy is a rapidly expanding area of research.

The development of T cells begins in the bone marrow, where they originate from progenitors that migrate to the thymus for maturation. Upon maturation in the thymus, T cells differentiate into two main lineages—conventional αβ T cells and unconventional γδ T cells—that can be distinguished by the expression of αβ or γδ T cell receptors, respectively. Conventional αβ T cells comprise various subsets, including cytotoxic T cells, helper T cells, regulatory T cells, natural killer T cells, and memory T cells, each with distinct functions in immune response. While the thymus output of naive T cells decreases with age, human naive T cells have a long intrinsic lifespan and turnover rate, making them a promising candidate for immunotherapy [8,9,10,11,12].

T cells have revolutionized cancer treatment, and their use as adoptive T cell therapy has shown great success for some cancers. However, the success of adoptive T cell therapy depends on various factors, such as the choice of T cell subset, target antigen, and immune evasion mechanisms employed by cancer cells. Additionally, the tumor microenvironment can influence T cell function and promote immune escape, highlighting the need for combination therapies targeting both the cancer cells and the immune system. Nevertheless, the potential of T cells to recognize and eliminate cancer cells has opened new avenues for cancer immunotherapy, providing hope for a future where cancer can be effectively treated and cured. This review provides an overview of T cells as a therapeutic tool for cancer, highlighting the advantages and future strategies for developing effective T cell cancer immunotherapy. We discuss the role of T cells in mediating immunity against cancer and recent advancements in T cell-based immunotherapy. We also highlight the challenges that remain, such as limited specificity, longevity, and toxicity, and outline future strategies for developing effective T cell cancer immunotherapy.

## 2. Antigenic Targets for T Cell Immunotherapy

Antigenic targets for T cell immunotherapy are crucial for the development of effective cancer treatments. The first human tumor antigen was discovered by Van der Bruggen et al. in 1991; it was named MZ2-E and was expressed in melanoma cells [7]. Consequently, cytotoxic T lymphocytes from melanoma patients recognized the melanoma cell line of HLA-A1 patients expressing the MZ2-E. Therefore, their group proposed that precise immunotherapy could be provided to HLA-A1+ melanoma patients expressing the MZ2-E antigen. Since then, various biochemical and genetic approaches have been developed to identify tumor antigens recognized by T cells. The biochemical approach involves eluting the HLA-peptide complex from tumor cells, followed by mass spectrometric sequencing to identify the antigenic peptides presented by the HLA molecules. In contrast, the genetic approach comprises two strategies: forward immunology and reverse immunology. The forward immunology approach involves cloning tumor-reactive T cells from patients and screening a cDNA library from patient-derived tumor cells to identify the antigens recognized by these T cells. On the other hand, the reverse immunology approach uses genome sequencing to identify mutations and predict potential HLA-binding antigens in the patient’s tumor. The peptide candidates are then screened for their ability to activate T cells in vitro [13]. T cells can target both HLA-restricted and HLA-non-restricted antigens found on the target cells. HLA-independent T cell immunotherapy targets antigens expressed on the surface of tumor cells. In contrast, HLA-dependent T cell immunotherapy targets antigens bound to HLA molecules on the surface of cells, known as the immunopeptidome.

### 2.1. Identification of Tumor Antigenic Epitopes for T Cell Immunotherapy

T cell immunotherapy is a promising approach for cancer treatment, but identifying tumor-reactive T cells and their target antigens is crucial for its success. There are two approaches to identifying these antigens: the indirect method, which involves screening candidate antigen targets of T cells from tumors in the laboratory, and the direct method, which involves isolating and identifying tumor-reactive T cells from tumor-infiltrating lymphocytes (TILs) or peripheral blood lymphocytes (PBLs) in patients. It is crucial to validate the reactivity of T cells towards tumors by studying the interactions between cancer cells and T cells in more depth. However, identifying T cells and their corresponding receptors with the ideal affinity for cancer antigens is challenging, as these antigens are often unmutated self-proteins expressed at higher levels in cancer cells than in normal tissues.

To generate antitumor T cells with optimal affinity of T cell receptor (TCR) to cancer-associated antigens, Obenaus et al. (2015) reported the potential of using antigen-negative humanized mice to generate T cells with a diverse human TCR repertoire and isolated the TCR specific to cancer antigens MAGE-A1 and NY-ESO-1, which have better affinity compared to human-derived TCRs [14]. Several other studies have attempted to enrich naturally occurring T cells from cancer patients by co-culturing TILs or PBLs with autologous tumors and expanding tumor-specific T cells before infusing them back into the patient [15,16,17,18]. However, the low frequency of T cells specific for these antigens makes it difficult to identify TCR T cells specific for cancer antigens. More recently, Arnaud et al. (2022) described a prediction method called NeoScreen that potentially identifies rare tumor antigens and their cognate TCR from TILs, enabling the selective expansion of antigen-specific T cells [19]. They reported that T cells transduced with target specific TCR identification via NeoScreen mediate tumor regression in preclinical models of cancer. Several T cell-based strategies have been developed to target tumor antigens that are well tolerated and show clinical promise in cancer patients. Overall, identifying tumor antigenic epitopes for T cell immunotherapy is crucial for the success of this promising approach to cancer treatment.

### 2.2. HLA Restricted T Cell-Based Immunotherapy

HLA dependent T cell-based immunotherapy employs the use of lymphocytes with minimally modified TCR or naturally occurring TCR directed against the tumor-specific antigen processed and presented by antigen-presenting cells via HLA molecules. This approach utilizes the ability of T cells to recognize the tumor antigens including cancer-associated or tumor-specific antigens presented by the antigen-presenting molecules on the cell surface. Many approaches have been employed for cancer immunotherapy in patients, such as endogenous T cells targeting undefined or defined antigens including cancer-associated or cancer-specific antigens.

#### 2.2.1. Endogenous T Cells Targeting Undefined Antigens

Endogenous T cell immunotherapy is a patient-specific type of therapy that involves using autologous tumor-infiltrating lymphocytes to treat tumors. In this type of T cell therapy, the tumor is surgically removed from the patient and TILs are dissected from the tumor. These TILs are then further expanded in the laboratory in the presence of cytokines, such as IL-2, which promotes the growth of T cells to reach a sufficient number for infusion back into the patient (Figure 1A). These naturally occurring T cells have the advantage of targeting multiple undefined antigens and can generate a nearly clonal or oligoclonal T cell response. In clinical trials, TILs have demonstrated success in a limited number of immunogenic cancers, such as melanoma and human papilloma-associated cancers. Moreover, TILs have a great amount of variability in their specificity and avidity for undefined antigens. In melanoma, TILS and PBL from patients that were stimulated in vitro using the patient’s tumor were found to kill the autologous tumor [20,21,22]. However, TIL infusion was associated with toxicity, and the most common side effects observed with TIL infusion included fever, hypotension, and flu-like symptoms, which were likely due to the high-dose interleukin-2 (IL-2) therapy that was used to support TIL expansion and activation [23]. Additionally, some patients experienced serious autoimmune toxicities, such as thyroiditis, colitis, and hepatitis, which were likely due to the reactivity of TILs against normal tissues in the body [24]. Overall, despite the variability in TIL specificity and avidity, the early TIL studies showed promise for treating certain types of advanced cancers but were limited by the significant toxicities associated with the therapy. Therefore, further research is needed to optimize TIL expansion, increase specificity and efficacy, decrease toxicity, and improve patient selection for this therapy.

Additionally, newer technologies such as genetic engineering of T cell receptors are being developed to enhance the efficacy of T cell therapy. T cells are being engineered for cancer immunotherapy through the introduction of T cell receptor (TCR) alpha and beta chains (Figure 1C) or chimeric antigen receptors (CARs) to target cancer cells (Figure 1D). The use of engineered TCRs is not without challenges, however, as the introduced TCRs can mispair with endogenous TCRs, potentially resulting in unpredictable specificity [25]. To minimize this risk, approaches that reduce mispairing and increase specificity are being developed. PBMCs obtained from either autologous or allogeneic donors through leukapheresis are used to introduce the transgene into the T cells through viruses using retro and lentiviral constructs or non-viral methods of delivery such as CRISPR-Cas9-based insertion of TCR [26]. The transduced T cells are then expanded in vitro under good manufacturing practices (GMP) before being infused back into the patient (Figure 1). The method of delivery for TCR transfer is critical, with viral methods being more effective than non-viral methods. However, challenges still exist in maximizing TCR expression, reducing mispairing between introduced and endogenous TCRs, and enhancing functional avidity of transduced T cells. Therefore, preclinical evaluation and screening of engineered T cells is crucial before proceeding to the clinical setting. With continued development and optimization, gene-engineered T cell therapy may hold promise as a highly targeted and effective approach to cancer treatment.

#### 2.2.2. T Cells Targeting Defined Cancer-Associated Antigens

Cancer-associated antigens (CAAs) are proteins that are overexpressed in tumor cells but rarely in normal cells [27,28]. Many CAAs have been identified through cDNA-based libraries, and targeting CAAs with T cells for adoptive cell therapy in patients expressing these antigens is a promising approach for treating advanced cancers. CDC27 is the first tumor-associated antigen identified that is well recognized by CD4+ TILS. Nagarsheth et al. (2021) reported the first in human, phase 1 clinical trial of T cells targeting CAA, HPV-16 E7 in metastatic human papilloma virus-associated epithelial cancers. The study displayed in vivo persistence and mediated robust tumor regression with an objective clinical response in 50% of patients [29]. T cells targeting tumor antigens in the tumor microenvironment have been reported in many cancers, and depending on whether the tumors are hot or cold, the frequency of these CAAs varies. However, chronic stimulation of tumor-associated antigen-specific T cells that bear high affinity TCR results in the elimination during negative selection in the thymus and hence loss of their anti-tumor ability. Nevertheless, not all high-affinity TCRs are eliminated by the negative selection and can be found in the body.

Other CAAs, such as cancer testis antigens (CTAs), are expressed in germline and placental trophoblast cells but are epigenetically silenced in normal cells. However, these CTAs have been found to be expressed in several cancers and elicit T cell-mediated responses. For example, T cells targeting NY-ESO1 is one of the successful immunotherapies used in a wide range of malignancies and has shown transient regression in more than 50% of patients with no off-target toxicity [30,31,32,33,34]. Similarly, CD4+ T cells engineered to target MHC class II displayed MAGE-A3 yielded positive responses in MAGE-A3-positive tumors with no toxicity but produced severe off-target toxicity when high-affinity TCR or MHC-I-restricted TCR was employed for target killing of cancer cells [35,36]. KK-LC-1 is also a CTA that is reported to be highly expressed in gastric cancer, breast cancer, lung cancer, and is currently being explored for safety and tolerability in clinical trials against KK-LC-1 highly-expressing tumors in clinical trials for many cancers (Table 1). The number of CTAs is more than 400 genes, and the list is ever-increasing in the CTA database [30]. Other CTAs such as GAGE, XAGE, BAGE and PAGE families, SSX1, SSX2 largely remain unexplored in many cancers, while targeting tumor-associated antigens such as MART-1, gp-100 with TCR T cell-directed therapy, also called tissue differentiation antigens, mediated tumor regression in 30% and 19% of patients, while at the same time targeting the normal tissues expressing cognate antigens in patients leads to off-target toxicity [37]. HER2, hTERT, and CEA are overexpressed in many epithelial cancers, but targeting them in T cell therapies carries a risk of toxicity. For instance, in a clinical trial for metastatic colorectal cancer patients, T cells that targeted CEA induced tumor regression but also caused severe transient inflammatory colitis [38]. Similarly, Oncofetal antigens which are CAAs include PSA, AFP, and WT1, and it is reported that T cells targeting WTI prevented AML relapse post-transplant with 100% relapse-free survival at a median of 44 months following infusion, showing promise for preventing AML recurrence [39].

#### 2.2.3. T Cells Targeting Defined Cancer-Specific Antigens

T cells targeting defined cancer-specific antigens (CSA) are highly tumor-specific and arise due to tumor-specific genomic alterations or from post-translational modifications. Neoantigens, also known as CSAs, are a well-characterized type of antigen that is expressed exclusively by cancer cells. They are generated by tumor-specific DNA alterations in the coding part of the genome, which creates a pool of new epitopes that are not present in somatic tissues. Neoantigens can be generated not only by DNA-specific alterations but also by other mechanisms such as viral infections, post-translational modifications, or gene rearrangements [40,41,42]. The advent of next-generation sequencing technology has enabled researchers to identify tumor-specific mutations and, in combination with binding prediction algorithms, has narrowed down the mutated peptides to potential immunogenic epitopes. Tumor mutational burden is a crude indicator of the number of cancer-specific antigens present in a tumor, and it is often the quality, rather than the quantity, of neoantigens that is critical for inducing an effective anti-tumor T cell response via the TCR.

The discovery of the first neoantigen dates back to 1988 when De Plaen and colleagues used a cDNA library screening to identify P91A, which differs from the normal gene by only one nucleotide and is recognized by cytolytic T lymphocytes [43]. In 1996, neoantigens derived from somatic mutations were reported in melanoma and renal cell carcinoma. With the rapid development of next-generation sequencing, the identification of mutated DNA sequences from normal and tumor cells became less labor-intensive. The steps in the identification of neoantigens involve the identification of non-synonymous mutations using whole-exome sequencing and RNA sequencing, selection of candidate neoantigens for immunogenicity prediction, and evaluation of immunogenicity of these neoantigens in a wet laboratory using different immunological assays. The mutations from the patient’s tumor can be identified, and binding to MHCs or immunogenicity of neoantigens can be predicted using high-throughput algorithmic platforms and validated in in vitro experimental setups (Figure 1B). Neoantigens that are immunogenic can induce an immune response via CD4+ and CD8+ T cell responses and generate neoantigen-specific clonotypes.

Comprehensive mapping of the mutational landscape of common forms of cancer is occurring at a rapid pace, leading to the identification of potential neoantigens, but strategies need to be developed to combine sequencing with mass spectrometry to yield true neoantigens that are relevant to T cell-based immunotherapy for the clinical benefit of cancer patients. The neoantigens are an important target for T cell therapy, but tumors can evade by depleting the neoantigens either at the DNA level by loss of copy number or at the RNA level by suppressing the RNA transcripts of neoantigens or at the epigenetic level via hypermethylation of genes coding neoantigens through post-translational mechanisms [44]. In summary, targeting defined cancer-specific antigens, particularly neoantigens, using T cell-based immunotherapy, has shown great promise in the treatment of cancer. However, further research is needed to improve the identification of neoantigens, as well as to overcome tumor escape mechanisms, to improve the efficacy of this approach.

##### Personalized and Shared Neoantigens as Targets of T Cell Therapy

Neoantigens can be classified as either personalized or shared. Personalized neoantigens are unique to each patient and have been found to drive anti-tumor CD8+ T cell responses in immune checkpoint blockade and TIL therapy, resulting in durable clinical benefits and improved progression-free survival in cancer patients [45,46,47,48]. In a study by Tran et al., a cancer-specific antigen called ERBB2IP was identified in a patient with metastatic cholangiocarcinoma. CD4+ T helper (Th1) cells from the patient were expanded and activated in vitro and then reinfused into the patient, resulting in significant tumor reduction and even tumor disappearance in some cases [18].

On the other hand, shared neoantigens are present in multiple types of cancer, such as KRAS, which is found in 60–70% of pancreatic adenocarcinomas and 20–30% of colorectal cancers [49,50]. A recent study reported that a shared neoantigen, tumor protein 53, in which arginine is replaced with histidine at position 175 (p53R175H), can be targeted by a highly specific TCR-mimic antibody that is HLA-A*0201-restricted p53R175H and can lyse tumor cells expressing the neoantigen [51]. Similarly, Kim et al. (2022) identified a pan-cancer epitope, collagen type VI α-3 (COL6A3), that was present in 30 tumor tissues derived from eight different cancer types. They also identified several TCRs that could recognize the shared neoantigen and efficiently eliminate tumors in preclinical models of cancer [52]. In a recent clinical trial, a 71-year-old woman with progressive pancreatic adenocarcinoma received a single infusion of engineered T cells expressing two allogeneic HLA-C*08:02-restricted TCRs targeting the KRASG12D neoantigen. The treatment resulted in an overall 72% objective partial tumor regression at 6 months, with the engineered T cells representing 2% of the total circulating T cells [53]. Targeting neoantigens has shown promise in leading to long-lasting clinical responses in many solid tumors, including epithelial cancers that account for more than 90% of cancer mortality rates in the United States [54].

Targeting shared neoantigens among solid tumors broadens the applicability of adoptive T cell therapy utilizing TCR T cells specific for common neoantigens. For example, out of 163 metastatic solid tumor patients screened for mutations by whole-exome sequencing, TP53 mutations were found in 78 patients [55]. Kim et al. (2022) reported a library of 39 TCRs that recognize tumor cells expressing TP53 mutations shared among more than 7% of patients with solid tumors, using in vitro and in vivo models. Moreover, the allogeneic use of TCR T cells targeting TP53 mutations in a chemo-refractory breast cancer patient showed that autologous PBLs engineered with R175H-TCR at 6 weeks post-treatment infiltrated the patient’s tumor and had acquired a central memory phenotype with stem-like features, suggesting the longevity of these cells. However, initial treatment with autologous PBLs engineered with R175H-TCR showed 55% objective regression that lasted only 6 months, with new cutaneous metastasis on the bilateral breast followed by death at 8 months due to other complications. Biopsies, WES, and RNAscope of progressing lesions confirmed the presence of the neoantigen TP53 on the tumor but loss of HLA-A*02:01 expression. Therefore, testing TCR T cell therapy directed against more than one shared neoantigen should be exploited, and off-target effects need to be evaluated in patient clinical trials to make the off-shelf T cell therapy available to advanced cancers with the mutated antigens.

Additionally, cancer cells can escape recognition by T cell-based immunotherapies by expressing the checkpoint molecules such as PD-L1, TIM3, LAG3 or downregulating the HLA expression which can further dampen T cell activity [56]. Thus, the expression of these molecules on cancer cells is an important factor to consider when designing T cell-based immunotherapies. Strategies to overcome PD-L1-mediated T cell suppression, such as the use of combination therapies that target multiple immune checkpoints, are being explored to enhance the efficacy of T cell-based immunotherapies in cancer treatment. Additionally, ongoing research is focused on identifying new targets and pathways to improve the effectiveness of T cell-based immunotherapies for cancer.

Moreover, using autologous T cells for T cell engineering has its own unique advantages and disadvantages over allogeneic T cell immunotherapies such as low risk of graft-versus-host disease (GVHD), since the T cells are derived from the patient’s own immune system. However, this therapy can be limited by the patient’s own T cell quality, quantity, and function. Additionally, it can be time-consuming and expensive to manufacture personalized T cell therapies for each patient. Allogeneic T cell immunotherapies, on the other hand, use T cells from a healthy donor, which can be modified to recognize and kill cancer cells (Figure 1C,D). The main advantage of allogeneic T cell immunotherapy is the potential for off-the-shelf availability and scalability, as a single donor can provide T cells for multiple patients. However, the major disadvantage of allogeneic T cell immunotherapy is the risk of GVHD, a serious complication where the donor T cells attack the recipient’s healthy cells. To address this risk, researchers are developing methods to mitigate the risk of GVHD, such as using gene-editing approaches to delete or downregulate genes that are involved in T cell activation and proliferation, such as CD52 and CD70, or engineering the T cells to express a suicide gene that can be triggered in case of GVHD [57,58,59]. Another approach is to use partially matched donors, which may reduce the risk of GVHD while still providing effective therapy.

### 2.3. Non-HLA Restricted T Cell-Based Immunotherapy

HLA-independent TCR-based T cell immunotherapy employs use of lymphocytes modified with TCR that are directed against the tumor antigens directly presented on the tumor cells. This approach utilizes a chimeric receptor introduced into the immune effector cells, such as T cells, natural killer cells or gamma delta (γδ) T cells to recognize tumor cell surface proteins and are commonly called as chimeric antigen receptor T or natural killer cells.

#### 2.3.1. Chimeric Antigen Receptor (CAR) T Cells Targeting Tumor-Defined Antigens

CAR T cell therapy is a type of immunotherapy that involves genetically engineering T cells to produce synthetic chimeric receptors that have an antigen-binding domain and a T cell-activating domain. This approach was first tested for HIV infection, but the CAR T cells showed no clinical improvement in HIV-infected individuals [60]. First-generation CARs were also tested in solid tumors targeting the MUC1 antigen but did not show longevity or persistence in clinical trials. The addition of a second co-stimulatory signal in CAR T cells, called second-generation CARs, was found to enhance the anti-tumor effect and persistence of CAR T cells in preclinical models of leukemia [61,62,63].

The team led by Carl June at the University of Pennsylvania involved in the early clinical trials of CAR T cell therapy played a critical role in the development of the first FDA-approved CAR T cell therapy, tisagenlecleucel. They designed CAR T cells targeting the B cell antigen CD19, coupled with CD137 and CD3-zeta, showing a low dose of these CAR T cells in a chronic lymphoid leukemia patient who showed a complete response after 3 weeks of treatment [64]. Similarly, they observed that CAR T cells with specificity to CD19 and a T cell-signaling molecule resulted in durable remission of acute lymphoblastic leukemia (ALL) in two pediatric patients with refractory and relapsed pre-B cell ALL [65]. The development of CAR T cells targeting the CD19 antigen resulted in a remission rate of close to 90% in r/r B-ALL, and 30–50% in chronic lymphocytic leukemia (CLL) and non-Hodgkin’s lymphoma in a phase I clinical trial. Consequently, researchers were able to circumvent the limits of conventional approaches by treating patients with CAR T-cells targeting the CD19 antigen, which stimulated recovery despite recurring malignancies [66,67]. CD19 CAR T cell treatment also has a significant anti-tumor effect in follicular lymphoma (FL), diffuse large B-cell lymphoma (DLBCL), mantle cell lymphoma (MCL), chronic lymphocytic leukemia (CLL), and multiple myeloma (MM) [68,69]. Tumor antigens evaluated in recent years for CAR T therapy include CD138, BCMA, CD123, LeY [70], and several others are in clinical trials (Table 1). Mesothelin CAR T cells have shown a positive response in a clinical study using patients with advanced mesothelioma and pancreatic cancer with no off-target effects [71]. Typically, the third-generation CARs are an expansion of the second generation and are composed of an antigen-binding unit, a spacer (hinge), a transmembrane domain and endodomain. Liu et al. (2017) evaluated the impact of the third generation of CAR T-cells that precisely target CD20 antigens, a glycosylated phosphoprotein expressed on the surface of activated B cells [72]. Patients with a wide range of B-lymphocyte-related lymphomas who were included in specialized clinical studies established positive clinical outcomes that were attributed to a well-executed therapy strategy. The concentration of antigen used in this therapy must meet a certain threshold for it to work properly; beyond that concentration, the treatment may have no impact on the progression of cancer in patients [73,74]. The fourth-generation CARs, also called ‘T cell redirected for universal cytokine-mediated killing’ (TRUCKs), are also based on the second-generation CARs with the addition of cytokine, IL-12, which is constitutively expressed, or have inducible expression promoting the tumor-killing via synergistic mechanisms. Similarly, fifth-generation CARs currently being explored have a truncated IL-2 receptor beta chain which upon activation triggers TCR and cytokine signaling via JAK-STAT/3/5, which drives the activation and proliferation of T cells.

Despite demonstrating promising results in preclinical and clinical trials, the efficacy of CAR T cell therapy may be limited by several unique characteristics of solid tumors. These characteristics include a deficiency in target antigens, heterogeneity of tumor antigens, inadequate trafficking and infiltration, and challenges imposed by the tumor microenvironment (TME), such as physical and metabolic barriers, the existence of soluble factors, and immunosuppressive cells. Several novel ways have been taken so far to overcome these obstacles, but the current methods are insufficient, and many more efforts are underway, such as combining CAR T cell technology with genome editing and adjuvant medicines, to discover a solution for more efficient therapeutics [75]. To date, a total of six CAR-T cell therapies directed against either the CD19 or B cell maturation antigens on B cells have been approved by FDA for the treatment of liquid tumors including lymphoma, leukemia, and multiple myeloma. These CAR T therapies are customized for each patient by collecting the blood and isolating T cells from patients, inserting the gene-encoding chimeric antigen receptor using vectors followed by selection, expansion in laboratory and infusion of these cells back to the patient (Figure 1D).

Despite the promising clinical results, current CAR T-cell therapies have also been associated with toxicities, but the nature and severity of these toxicities differ from those seen in early TIL studies. The most common toxicity associated with CAR T-cell therapy is cytokine release syndrome (CRS), which is caused by the release of large amounts of inflammatory cytokines as the CAR T-cells attack cancer cells [76,77]. CRS can cause fever, hypotension, and other flu-like symptoms, and can be severe or even life-threatening in some cases. Another common toxicity associated with CAR T-cell therapy is neurotoxicity, which can cause confusion, seizures, and other neurological symptoms [78]. However, these toxicities can often be managed with supportive care and, in some cases, with the use of immunosuppressive agents.

A potential alternative for CAR T cell therapy is CAR NK cell therapy by engineering natural killer cells, which are a type of innate immune cell that can recognize and eliminate abnormal cells including tumor cells. The advantages of using NK cells for immunotherapy include their ability to rapidly recognize and eliminate cancer cells, as well as their lack of requirement for prior sensitization, which can make them a more practical and accessible option than other types of immune cells. Additionally, NK cells have been shown to have a lower risk of causing graft-versus-host disease (GVHD) than T cells, which can be an important consideration in allogeneic transplant settings [79]. However, there are also some limitations and challenges associated with the use of NK cells in immunotherapy. For example, the efficacy of NK cell therapy can be affected by the immunosuppressive tumor microenvironment, which can impair NK cell function and limit their ability to eliminate cancer cells. Additionally, there is a need to identify reliable and effective methods for expanding and activating NK cells ex vivo for use in immunotherapy. Despite these challenges, ongoing research is focused on developing new strategies to overcome these limitations and improve the efficacy of NK cell-based immunotherapies [80]. These strategies include the use of combination therapies that target multiple immune pathways, the development of novel methods for NK cell expansion and activation, and the exploration of targeted delivery approaches to enhance NK cell infiltration and activity in the tumor microenvironment.

Similarly, in recent years mucosal-associated invariant T cells (MAIT cells) have been used as a potential source for CAR T cell therapy [81]. MAITs are a subset of T cells that recognize antigens presented by a non-classical major histocompatibility complex molecule, MR1. These cells are not HLA-restricted and are not expected to induce GVHD, which gives them huge potential as a source for off-the-shelf immunotherapy. Preclinical studies have shown that MAIT cells can be engineered to express CARs that target cancer cells and that these CAR MAIT cells are able to recognize and kill cancer cells in vitro and in vivo [82]. Additionally, MAIT cells have shown promise in treating solid tumors, which are often resistant to traditional CAR T cell therapy. However, there are still challenges to be addressed in the development of CAR MAIT cells. One of the main challenges is the limited understanding of the biology and function of MAIT cells, which makes it difficult to optimize the engineering and expansion of CAR MAIT cells. Additionally, the development of CAR MAIT cells requires the identification of suitable tumor-associated antigens that can be targeted by the CAR. Nonetheless, the potential of CAR MAIT cells as a novel immunotherapy approach for cancer treatment is an active area of research, and ongoing studies are focused on addressing these challenges and optimizing the use of these cells in the clinic.

#### 2.3.2. Gamma Delta (γδ) T Cells as Weapons of Non-HLA-Restricted Immunotherapy

Gamma delta T cells (γδ) are a unique subgroup of T cells that make up 0.5–5% of all T cells and express T cell receptors composed of γ and δ chains. These cells have the ability to recognize non-HLA-restricted tumor antigens and can release antitumor cytokines, making them a promising candidate for T cell immunotherapy. In humans, there are four subgroups of γδ T cells based on the type of δ TCR chains (δ1, δ2, δ3, δ5) paired with seven different Vγ TCR chains (Vγ2, Vγ3, Vγ4, Vγ5, Vγ8, Vγ9). Recent research has shown that, depending on the subtype, γδ T cells can act as effector T cells playing a role in anti-tumor activity or promote the growth of the tumor by acting as regulatory cells, indicating a dual effect in cancer immunotherapy. Effector γδ T cells release granzyme B, perforin, IFN- γ and TNF-a, displaying a strong tumor-cell-killing activity. High infiltration of effector γδ T cells has been reported to improve the clinical outcome of patients with different malignancies [83,84,85,86,87]. Previous early clinical trials on the therapeutic potential of γδ T cells showed limited clinical benefit in renal, multiple myeloma, and non-small lung cancer progression, although they were well tolerated and safe [88,89,90]. More recent studies have revealed that γδ T cells, which express PD-1, are the primary effectors of immunotherapy in DNA mismatch repair-deficiency tumors that possess HLA class I defects [91]. This finding highlights the potential of these cells in HLA-non-restricted T cell immunotherapy, either alone or in conjunction with immune checkpoint blockade. Currently, various methods are being assessed to harness the potential of γδ T cells, including the development of γδ CAR-T cells (Table 1) to enhance therapeutic efficacy and minimize the risk of toxicity compared to αβ CAR-T cells [92,93,94,95,96,97]. Additionally, antibody-based approaches utilizing γδ TCR-specific engagers directed against antigens such as Her2 and CD123 have shown promise in killing tumor cells, as they have bispecific nanobody approaches targeting CD40, epidermal growth factor receptor in combination with γδ TCR T cells [98,99]. In summary, γδ T cells hold promise as a foundation for modifying the targeting of different cancer antigens and translating them into clinical benefits for patients. Further research and clinical trials are needed to fully realize their potential as a non-HLA-restricted immunotherapy.

## 3. Conclusions and Future Perspectives

In conclusion, cancer immunotherapies have shown great potential in harnessing the patient’s own immune system to fight cancer. The discovery of antitumor T cells infiltrating tumors has opened doors for T cell immunotherapy targeting tumor antigens. Over the years, T cell-based immunotherapy targeting antigens has been the focus of much research and development, with different approaches being employed for the treatment of cancer. This includes the use of autologous T cells isolated from the patient tumor or engineering the autologous or allogeneic donor T or NK cells to express tumor-specific T cell receptors (TCRs) or chimeric antigen receptors (CARs) that specifically target cancer cells. While HLA-restricted T cell therapy and non-restricted CAR T cell therapies have shown great potential in clinical trials, challenges such as the development of resistance to therapy and the identification of new targets for T cell-based immunotherapy remain. Additionally, off-target toxicity and tumor heterogeneity are concerns that need to be addressed. Identifying multiple targeted antigens and developing T cell-based therapies that express multiple antigen-specific CARs or TCRs specific for or shared among patients could overcome the tumor heterogeneity and increase effectiveness. Furthermore, part of a tumor may escape the immune response by upregulating check-point molecules that suppresses the effector T cell function or by losing HLA expression, causing a deficiency in the antigen processing and presentation machinery. Combination therapies targeting tumor antigens along with checkpoint inhibitors or use of alternative cell sources such as NK cells, MIAT cells, and γδ T cells may offer solutions. It is important to vigilantly monitor the safety associated with the T cell-based immunotherapies in preclinical settings before using them for the clinical benefit. Additionally, more research should be done employing the NK cells, MIAT cells and γδ T cells as a potential source for off-the-shelf antigen-directed therapies for cancer. Continued research, development, and refinement of these approaches could make it a standard treatment option for a wide range of cancers, improving patient outcomes and survival rates. While there is still much work to be done, the potential benefits of T cell-based immunotherapy are clear, and the field is poised to make significant strides towards more effective and safe cancer treatments.

## Figures and Tables

**Figure 1 vaccines-11-00835-f001:**
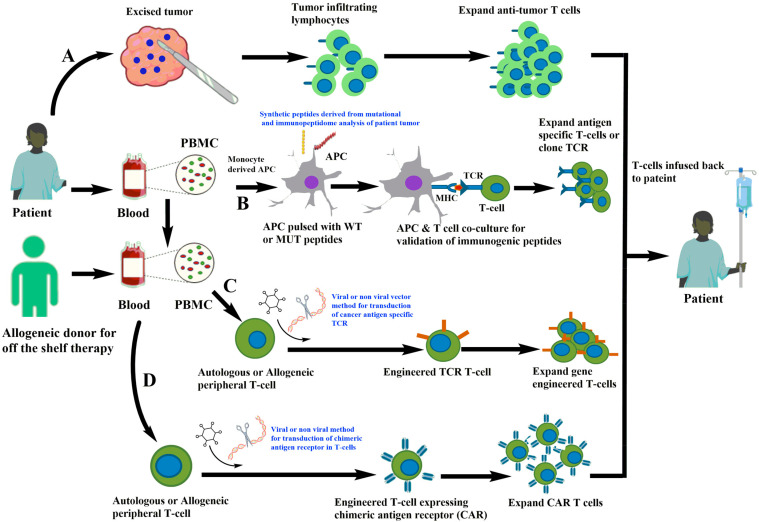
Different strategies of T cell immunotherapies. (**A**) Isolation of tumor-infiltrating lymphocytes from patient tumor followed by expansion of anti-tumor T cells in vitro and infusion back to the patients. (**B**) Generation of antigen-presenting cells from monocytes derived from patient PBMC and loaded with potential antigens derived from the mutational and immunopeptidome analysis of patient tumor. The antigen-loaded APC are co-cultured with autologous T cells that are screened for anti-tumor T cells and further expanded. (**C**) TCR is cloned and used to gene-engineer TCR in fresh autologous or allogeneic T cells for expansion in laboratory conditions and infusion back to the patient. (**D**) Isolation of T cells from patient or allogeneic donor and gene-editing approaches for engineering of chimeric antigen receptors targeting cancer cells.

**Table 1 vaccines-11-00835-t001:** List of HLA-restricted and non-HLA restricted T cell cancer immunotherapies in clinical trials targeting many different antigens.

T Cell Therapies	Antigen/Type	Disease	HLA Subtype	Status	Reference/Clinical Trial Identifier
TCR-T	KK-LC-1/cancer testis antigen Kita-kyushu Lung Cancer Antigen 1	Gastric, breast, cervical, lung, other KK-LC-1 and other positive epithelial cancers	HLA-A01:01	Phase I	NCT05483491 NCT05035407 NCT03778814
E7 TCR cells	HPV-16 E7	Human papillomavirus (HPV)-16+ cancers (cervical, vulvar, vaginal, penile, anal, and oropharyngeal cancers	HLA-A02:01	phase I/II	NCT02858310
TCR-transduced CD4+ and CD8+ T-cells FH-MCVA2TCR	MCPyV T	Merkel cell cancer	HLA-A02	phase I/II trial	NCT03747484
LMP2 Antigen-specific TCR T cells	LMP2	Recurrent and metastatic nasopharyngeal carcinoma	HLA-A2, HLA-A11, HLA-A24	Phase 1	NCT03925896
NY-ESO-1 TCR-T	NY-ESO-1	Advanced soft tissue sarcoma	HLA-A 02/01	Recruiting	NCT05620693
TCR-T therapy	KRAS G12V or G12D	Pancreatic cancer	HLA-A*11:01	Phase 1	NCT05438667
EBV-Specific Anti-PD1 TCR-T Cells	EBV antigen	Head and neck squamous cell carcinoma	Unknown	Phase I/II trial	NCT04139057
Tumor-specific TCR-T cells	Patient specific	Solid tumor	Unknown	Phase 1	NCT03891706
HPV TCR-T	HPV E6	HPV-positive head and neck carcinoma or cervical cancer	Unknown	Phase 1	NCT03578406
TCR-transduced T cells	TSA-reactive TCR-engineered T cells	Malignant epithelial neoplasms	Unknown	Phase 1	NCT04520711
CRTE7A2-01 TCR-T Cell	HPV16	Cervical, anal, head and neck cancers	HLA-A*02:01	Phase 1	NCT05122221
MC2 TCR T cells	MAGE-C2	Melanoma melanoma, uveal head and neck cancer	HLA-A2	Phase 1/2	NCT04729543
Neoantigen specific TCR-T cells	Patient specific	Gynecologic cancer colorectal, pancreatic cancer, non-small cell lung cancer, cholangiocarcinoma, ovarian cancer, ovary neoplasm, squamous cell lung cancer, adenocarcinoma of lung, adenosquamous cell lung cancer	Unknown	Phase I/II	NCT05292859
KSH01-TCR-T cells	Unknown	Refractory/recurrent solid tumors	HLA-A*02	Early Phase 1	NCT05539833
Neoantigen specific TCR-T cell	KRAS G12D, KRAS G12V, TP53 R175H, TP53 R248W, TP53 Y220C, EGFR E746-A750del	Gynecologic, colorectal, pancreatic, non-small cell lung cancer, cholangiocarcinoma ovarian, endometrial cancer, ovary neoplasm, squamous cell lung cancer, adenocarcinoma of lung, adenosquamous cell lung cancer	HLA-A*11:01 HLA-C*08:02 HLA-A*11:01 HLA-C*01:02 HLA-A*02:01, HLA-DRB1*13:01, HLA-A*68:01, HLA-DRB3*02:02, HLA-DPA1*02:01, DPB1*01:01	Phase I/II	NCT05194735
KK-LC-1 TCR T	KK-LC-1	Gastric, breast, cervical, lung cancer	HLA-A01:01	Phase 1	NCT05035407
FH-TCR-Tᴍsʟɴ cells	Mesothelin	Metastatic pancreatic ductal adenocarcinoma	HLA-A*02:01	Phase 1	NCT04809766
NY-ESO-1 T cell receptor (TCR) engineered T cells	NY-ESO-1	Ovarian, Fallopian tube, or primary peritoneal cancer	HLA-A*0201, HLA-DP*04	Phase 1	NCT03691376
fhB7H3.CAR-Ts	B7H3	Ovarian cancer	Non-HLA	Phase I/II	NCT05211557
Anti-MUC1 CAR-T cells	MUC1	Advanced esophageal cancer	Non-HLA	Phase I/II	NCT03706326
Anti-CEA-CAR T	CEA	Colorectal cancer	Non-HLA	Phase I	NCT04513431
anti-MESO CAR-T cells	Mesothelin	Ovarian cancer	Non-HLA	Phase 1/2	NCT03916679
EpCAM CAR-T cells	EpCAM	Advanced solid tumors	Non-HLA	Phase 1	NCT02915445
CAR-T cells	PSCA, MUC1, TGFβ, HER2, Mesothelin, Lewis-Y, GPC3, AXL, EGFR, Claudin18.2, or B7-H3	Lung cancer	Non-HLA	Phase 1	NCT03198052
CD70-targeted CAR-T Therapy	CD70	Advanced renal cancer	Non-HLA	Phase 1	NCT05420519
Dual Targeted CAR T-cells	CD20/CD22	Relapsed or refractory lymphoid malignancies	Non-HLA	Phase 1	NCT04283006
MOv19-BBz CAR T Cells	Alpha folate receptor	Recurrent high-grade serous ovarian, Fallopian tube, or primary peritoneal cancer	Non-HLA	Phase 1	NCT03585764
U87 CAR-T	U87	Pancreatic cancer	Non-HLA	Phase 1	NCT05605197
LMP1 CAR T-cells	LMP1	Hematological malignancies	Non-HLA	Phase 1	NCT04657965
IM83 CAR-T Cells	IM83	Liver cancer	Non-HLA	Phase 1	NCT05123209
CAR T cells	ICAM-1	Relapsed/refractory thyroid cancer	Non-HLA	Phase 1	NCT04420754
anti- MESO CAR-T cells	Mesothelin	Ovarian cancer	Non-HLA	Phase 1	NCT03814447
huCART-meso cells	Mesothelin	Pancreatic cancer	Non-HLA	Phase 1	NCT03323944
Anti-CD33-CAR-transduced T cells	CD33	Myeloid malignancies	Non-HLA	Phase 1/2	NCT02958397
Anti-CD7 CAR-T	CD7	Hematological malignancies	Non-HLA	Phase 2	NCT05454241
B4T2-001 Autologous CAR T cells	BT-001	Advanced solid tumor	Non-HLA	Phase 1	NCT05621486
CEA CAR-T cells	CEA	Solid tumor, lung cancer, colorectal cancer, liver cancer, pancreatic cancer, gastric cancer, breast cancer,	Non-HLA	Phase 1/2	NCT04348643
PSCA- CAR T cells	PSCA	Metastatic castration-resistant prostate cancer	Non-HLA	Phase 1	NCT03873805
PRGN-3005 Ultra CAR-T cells	BRCA	Advanced, recurrent platinum-resistant ovarian, Fallopian tube or primary peritoneal cancer	Non-HLA	Phase 1	NCT03907527
P-PSMA-101 CAR-T cells	prostate-specific membrane antigen (PSMA)	Metastatic castration-resistant prostate cancer and advanced salivary gland cancers	Non-HLA	Phase 1	NCT04249947
BPX-601 CAR-T cells	PSCA	Metastatic castration-resistant prostate cancer	Non-HLA	Phase 1/2	NCT02744287
Allogeneic γδ T cells in combination with standard drugs	Unknown	Neuroblastoma	Non-HLA	Phase 1	NCT05400603
CAR γδ T-cells	NKG2DL	Malignant cancer	Non-HLA	Phase 1	NCT05302037
CAR γδ T-cells	CD7	Malignant cancer	Non-HLA	Phase 1	NCT04702841
Autologous and allogeneic γδ T-cells	Unknown	Glioblastoma	Non-HLA	Phase 1/2	NCT05664243
Allogeneic γδ T-cells	Unknown	Non-Hodgkin’s lymphoma, peripheral T cell lymphoma	Non-HLA	Phase 1	NCT04696705

## Data Availability

Not applicable.

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
