# Peer review of "T Cell Based Immunotherapy for Cancer: Approaches and Strategies"

_vaccines, 2023, doi:10.3390/vaccines11040835_

Round 1
Reviewer 1 Report
Want et al. present a timely review of a complex area. This reviewer has no comments on the content of the existing manuscripts but I have these suggestions:
- A brief comment on the place of NK cells in immunotherapy – although they would formally be ‘beyond the scope’ of a T cell review – they should comment on potential advantages and disadvantages of an NK approach
- The authors should write a paragraph of the pros and cons of allogeneic vs autologous T cell immunotherapies – ie risk of GVHD etc and how they are being addressed
- A comment on ‘toxicities’. Were there toxicities with the early TIL studies for example compared to the current CART trials
- A comment on invariant T cells (i.e. MAIT) would be worthwhile as there has been some recent studies on chimeric MAITs
Author Response
We would like to express our gratitude for taking the time to review our manuscript. Your valuable feedback has helped us to identify the areas of improvement and enhance the overall quality and clarity of our paper. We sincerely appreciate your time and effort in providing us with insightful and constructive comments. Your expertise and insights have been invaluable for shaping our manuscript. Thank you again for your valuable feedback.
Comment #1. A brief comment on the place of NK cells in immunotherapy – although they would formally be ‘beyond the scope’ of a T cell review – they should comment on potential advantages and disadvantages of an NK approach
Response: We appreciate your valuable feedback on our manuscript. We have added a paragraph on the place of NK in immunotherapy in line 431-449 as A potential alternative for CAR T cell therapy is CAR NK cell therapy by engineering natural killer cells, which are a type of innate immune cell that can recognize and eliminate abnormal cells including tumor cells. The advantages of using NK cells for immunotherapy include their ability to rapidly recognize and eliminate cancer cells, as well as their lack of requirement for prior sensitization, which can make them a more practical and accessible option than other types of immune cells. Additionally, NK cells have been shown to have a lower risk of causing graft-versus-host disease (GVHD) than T cells, which can be an important consideration in allogeneic transplant settings. However, there are also some limitations and challenges associated with the use of NK cells in immunotherapy. For example, the efficacy of NK cell therapy can be affected by the immunosuppressive tumor microenvironment, which can impair NK cell function and limit their ability to eliminate cancer cells. Additionally, there is a need to identify reliable and effective methods for expanding and activating NK cells ex vivo for use in immunotherapy. Despite these challenges, ongoing research is focused on developing new strategies to overcome these limitations and improve the efficacy of NK cell-based immunotherapies. These strategies include the use of combination therapies that target multiple immune pathways, the development of novel methods for NK cell expansion and activation, and the exploration of targeted delivery approaches to enhance NK cell infiltration and activity in the tumor microenvironment.
Comment #2 The authors should write a paragraph of the pros and cons of allogeneic vs autologous T cell immunotherapies – ie risk of GVHD etc and how they are being addressed
Response: Thank your kind feedback. We have now included a paragraph on pros and cons of allogeneic vs autologous T cell immunotherapies in lines 334-350 as Moreover, using autologous T cells for T cell engineering have their own unique advantages and disadvantages over allogeneic T cell immunotherapies such as low risk of graft-versus-host disease (GVHD) since the T cells are derived from the patient's own immune system. However, this therapy can be limited by the patient's own T cell quality, quantity, and function. Additionally, it can be time-consuming and expensive to manufacture personalized T cell therapies for each patient. Allogeneic T cell immunotherapies, on the other hand, use T cells from a healthy donor, which can be modified to recognize and kill cancer cells. The main advantage of allogeneic T cell immunotherapy is the potential for off-the-shelf availability and scalability, as a single donor can provide T cells for multiple patients. However, the major disadvantage of allogeneic T cell immunotherapy is the risk of GVHD, a serious complication where the donor T cells attack the recipient's healthy cells. To address this risk, researchers are developing methods to mitigate the risk of GVHD is to use gene editing approaches to delete or downregulate genes that are involved in T cell activation and proliferation, such as CD52 and CD70, or engineering the T cells to express a suicide gene that can be triggered in case of GVHD. Another approach is to use partially matched donors, which may reduce the risk of GVHD while still providing effective therapy.
Comment #3: A comment on ‘toxicities’. Were there toxicities with the early TIL studies for example compared to the current CART trials
Response: Thank you for the valuable point. We have added toxicities of TILs and CAR T cells in lines 139-144 and 420-429 respectively as However, TIL infusion was associated with toxicity and the most common side effects observed with TIL infusion included fever, hypotension, and flu-like symptoms, which were likely due to the high-dose interleukin-2 (IL-2) therapy that was used to support TIL expansion and activation. Additionally, some patients experienced serious autoimmune toxicities, such as thyroiditis, colitis, and hepatitis, which were likely due to the reactivity of TILs against normal tissues in the body. Overall, despite the variability in TIL specificity and avidity, the early TIL studies showed promise for treating certain types of advanced cancers, but were limited by the significant toxicities associated with the therapy. Therefore, further research is needed to optimize TIL expansion, increase specificity and efficacy, decrease toxicity, and improve patient selection for this therapy.
in line 420-429 as Despite the promising clinical results, current CAR T-cell therapies have also been associated with toxicities, but the nature and severity of these toxicities differ from those seen in early TIL studies. The most common toxicity associated with CAR T-cell therapy is cytokine release syndrome (CRS), which is caused by the release of large amounts of inflammatory cytokines as the CAR T-cells attack cancer cells. CRS can cause fever, hypotension, and other flu-like symptoms, and can be severe or even life-threatening in some cases. Another common toxicity associated with CAR T-cell therapy is neurotoxicity, which can cause confusion, seizures, and other neurological symptoms. However, these toxicities can often be managed with supportive care and, in some cases, with the use of immunosuppressive agents.
Comment #4: A comment on invariant T cells (i.e. MAIT) would be worthwhile as there has been some recent studies on chimeric MAITs
Response: Thank you again for this valuable suggestion. We have included a paragraph on MAIT and included some studies on MAIT as source of CAR T cells in lines 451-467 as Similarly, in recent years mucosal-associated invariant T cells (MAIT cells) has been used as a potential source for CAR T cell therapy[81]. MAIT’s are a subset of T cells that recognize antigens presented by a non-classical major histocompatibility complex (MHC) molecule, MR1. These cells are not HLA restricted and are not expected to induce GVHD, thereby having huge potential as a source for off the shelf immunotherapy. Preclinical studies have shown that MAIT cells can be engineered to express CARs that target cancer cells, and that these CAR MAIT cells are able to recognize and kill cancer cells in vitro and in vivo. Additionally, MAIT cells have shown promise in treating solid tumors, which are often resistant to traditional CAR T cell therapy. However, there are still challenges to be addressed in the development of CAR MAIT cells. One of the main challenges is the limited understanding of the biology and function of MAIT cells, which makes it difficult to optimize the engineering and expansion of CAR MAIT cells. Additionally, the development of CAR MAIT cells requires the identification of suitable tumor-associated antigens that can be targeted by the CAR. Nonetheless, the potential of CAR MAIT cells as a novel immunotherapy approach for cancer treatment is an active area of research, and ongoing studies are focused on addressing these challenges and optimizing the use of these cells in the clinic.
Reviewer 2 Report
The review gives a general historical overview on T-cell based immunotherapeutic approaches.
It is well written and clear.
It is indeed very didactic and helpful for people who are knew in the filed but maybe too much general for experts in the field.
- I would divide the Table making one for TCR and another one for CARs;
- The figure is not representative of all the potential sources (allogeneic, off the shelf) and it is missing also the non-viral gene transfer approaches, which are indeed becoming among the most used ones in parallel to viral vectors)
- In the references on CARs, I would add some more seminal works (i.e. by the Upenn team) published in high impact journals (maybe replacing the current ones between 58-60).
Author Response
We would like to express our gratitude to the reviewer for taking the time to review our manuscript. Your valuable feedback has helped us to identify the areas of improvement and enhanced the overall quality of the manuscript. We have carefully reviewed each of your comments and have made the necessary changes to address the comments you have raised.
Comment #1: I would divide the Table making one for TCR and another one for CARs;
Response: Thank you for this observation. We thought same before submitting the original manuscript but we apologize that keeping in view the journal guidelines, we included TCR and CARs in single table with first part of the table showing HLA restricted T cell therapies and second part consisting of HLA non-restricted therapies.
Comment #2: The figure is not representative of all the potential sources (allogeneic, off the shelf) and it is missing also the non-viral gene transfer approaches, which are indeed becoming among the most used ones in parallel to viral vectors).
Response: Thank you for pointing out this. We agree potential sources (allogeneic, off the shelf) and non-viral gene transfer approaches is missing in Figure 1. We have now modified the figure 1 and included the allogeneic donor and non-viral approaches in the updated figure.
Comment #3: In the references on CARs, I would add some more seminal works (i.e. by the Upenn team) publishure, the UPENN team, led by Dr. Carl June, has been a leader in the development of CAR T cell therapy for cancer.
Response: Thank you for insightful comment on our manuscript. We have also added three important referenes of UPENN team published in NEJM and Sci Translational Medicine and have added seminal works by the UPENN team in lines 337-334 as The team lead by Carl June at the University of Pennsylvania involved in the early clinical trials of CAR T cell therapy played a critical role in the development of the first FDA-approved CAR T cell therapy, tisagenlecleucel. They designed CAR T targeting the B cell antigen, coupled with CD137 and CD3-zeta showing low dose of these CAR T cells in chronic lymphoid leukemia patient showed complete response after 3 weeks of treatment. Similarly, they observed CAR T cells with specificity to CD19 and a T cell signaling molecule resulted in durable remission of acute lymphoblastic leukemia (ALL) in two pediatric patients with refractory and relapsed pre-B cell ALL. We have also added three important referenes of UPENN team published in NEJM and Sci Translational Medicine.
- 64. Porter, D.L.; Levine, B.L.; Kalos, M.; Bagg, A.; June, C.H. Chimeric antigen receptor-modified T cells in chronic lymphoid leukemia. N Engl J Med 2011, 365, 725-733, doi:10.1056/NEJMoa1103849
- Grupp, S.A.; Kalos, M.; Barrett, D.; Aplenc, R.; Porter, D.L.; Rheingold, S.R.; Teachey, D.T.; Chew, A.; Hauck, B.; Wright, J.F.; et al. Chimeric antigen receptor-modified T cells for acute lymphoid leukemia. N Engl J Med 2013, 368, 1509-1518, doi:10.1056/NEJMoa1215134.
- Kalos, M.; Levine, B.L.; Porter, D.L.; Katz, S.; Grupp, S.A.; Bagg, A.; June, C.H. T cells with chimeric antigen receptors have potent antitumor effects and can establish memory in patients with advanced leukemia. Sci Transl Med 2011, 3, 95ra73, doi:10.1126/scitranslmed.3002842.
Reviewer 3 Report
In this research, the authors reviewed the “T cells-based Immunotherapy for cancer: approaches and strat-2 egies” In my opinion, the current stage of this paper could meet the requirements of Vaccines after major revisions.
My comments are as details:
1. In this review, the author focused on the T cell therapy. But, how other immune pathways such as PD-L1 affect the T cells-based Immunotherapy for cancer were neglected. In my opinion, this part should be added. Some references should be added to this part including 10.1016/j.ijbiomac.2022.10.167 and 10.1016/j.molcel.2018.07.030.
2. The conclusion part was too plain. An in-depth outlook or conclusion should be added.
3. The quality of the figures especially Figure 1 still could be improved if possible.
4. Currently, some efforts were made to increase the infiltration and activity of T cell therapy. In my opinion, this part should be added. Some references should be added to this part including 10.1002/adma.202206121 and 10.1016/j.apsb.2022.07.023.
5. Some minor mistakes exist in this review. The authors should carefully check and revise it.
Author Response
We would like to express our gratitude for taking the time to review our manuscript. Your valuable feedback has helped us to identify the areas of improvement and enhance the overall quality and clarity of our paper. We sincerely appreciate your time and effort in providing us with insightful and constructive comments. Your expertise and insights have been invaluable for shaping our manuscript. Thank you again for your valuable feedback.
Comment #1: In this review, the author focused on the T cell therapy. But, how other immune pathways such as PD-L1 affect the T cells-based Immunotherapy for cancer were neglected. In my opinion, this part should be added. Some references should be added to this part including 10.1016/j.ijbiomac.2022.10.167 and 10.1016/j.molcel.2018.07.030.
Response: Thank you for suggestion. Although this is beyond the scope of this manuscript and we apologize we found the references as suggested are irrelevant and have included few lines with relevant references in lines 324-332 as Additionally, cancer cells can escape recognition by T cell-based immunotherapies by expressing the check point molecules such as PD-L1, TIM3, LAG3 or down regulating the HLA expression which can further dampen T cell activity. Thus, the expression of these molecules on cancer cells is an important factor to consider when designing T cell-based immunotherapies. Strategies to overcome PD-L1-mediated T cell suppression, such as the use of combination therapies that target multiple immune checkpoints, are being explored to enhance the efficacy of T cell-based immunotherapies in cancer treatment. Additionally, ongoing research is focused on identifying new targets and pathways to improve the effectiveness of T cell-based immunotherapies for cancer.
Comment #2: The conclusion part was too plain. An in-depth outlook or conclusion should be added.
Response: Thank you for insightful comment on our manuscript. We greatly appreciate your suggestion for in-depth conclusion, and we have now made changes in the conclusion in lines 506-530 as In conclusion, cancer immunotherapies have shown great potential in harnessing the patient's own immune system to fight cancer. The discovery of antitumor T cells infiltrating the tumor has opened doors for T cell immunotherapy targeting tumor antigens. Over the years, T cell-based immunotherapy targeting antigens have been the focus of much research and development, with different approaches being employed for the treatment of cancer. This includes the use of autologous T cells isolated from the patient tumor or engineering the autologous or allogeneic donor T or NK cells cells to express tumor-specific T cell receptors (TCRs) or chimeric antigen receptors (CARs) that specifically target cancer cells. While, HLA restricted T cell therapy and non-restricted CAR T cell therapies have shown great potential in clinical trials, challenges such as the development of resistance to therapy and the identification of new targets for T cell-based immunotherapy remain.Additionally, off-target toxicity and tumor heterogeneity are concerns that need be addressed. Identifying multiple targeted antigens and developing T cell-based therapies that express multiple antigen-specific CARs or TCRs specific or shared among patients could over come the tumor heterogeneity and increase effectiveness. Furthermore, a part of tumor may escape the immune response by upregulating check point molecules that suppresses the effector T cell function or by losing HLA expression, causing a deficiency in the antigen processing and presentation machinery. Combination therapies targeting tumor antigens along with check point inhibitors or use of alternative cell sources such as NK cells, MIAT cells, and γδ T cells may offer solutions. It is important to vigilantly monitor the safety associated with the T cell-based immunotherapies in preclinical settings before using them for the clinical benefit. Additionally, more research should be done employing the NK cells, MIAT cells and γδ T cells as a potential source for off the shelf antigen directed therapies for cancer. Continued research, development, and refinement of these approaches could make it a standard treatment option for a wide range of cancers, improving patient outcomes and survival rates. While there is still much work to be done, the potential benefits of T cell-based immunotherapy are clear, and the field is poised to make significant strides towards more effective and safe cancer treatments.
Comment #3: The quality of the figures especially Figure 1 still could be improved if possible.
Response: Thank you for this observation. we have now modified and improved the quality of the figure 1 in the manuscript.
Comment #4: Currently, some efforts were made to increase the infiltration and activity of T cell therapy. In my opinion, this part should be added. Some references should be added to this part including 10.1002/adma.202206121 and 10.1016/j.apsb.2022.07.023.
Response: We apologize the work done by the Jianliang et al as mentioned in the two references cannot be added since it is beyond the scope of our manuscript.
Comment #5. Some minor mistakes exist in this review. The authors should carefully check and revise it.
Response: Thank you for this observation. We have carefully revised for any minor mistakes in the review.
Round 2
Reviewer 3 Report
The current version of this manuscript could be accepted.